# Social Media Use and Its Association with Mental Health and Internet Addiction among Portuguese Higher Education Students during COVID-19 Confinement

**DOI:** 10.3390/ijerph20010664

**Published:** 2022-12-30

**Authors:** Ana Paula Oliveira, Joana Rita Nobre, Henrique Luis, Luis Soares Luis, Lara Guedes Pinho, Núria Albacar-Riobóo, Carlos Sequeira

**Affiliations:** 1Health School, Polytechnic Institute of Portalegre, 7300-555 Portalegre, Portugal; 2Faculty of Nursing, University of Rovira e Virgili, 43003 Tarragona, Spain; 3Unidade de Investigação em Ciências Orais e Biomédicas (UICOB), Faculdade de Medicina Dentária, Universidade de Lisboa, Rua Teresa Ambrósio, 1600-277 Lisbon, Portugal; 4Center for Innovative Care and Health Technology (ciTechcare), Polytechnic of Leiria, 2410-541 Leiria, Portugal; 5School of Health Sciences, Polytechnic of Leiria, 2410-541 Leiria, Portugal; 6Nursing School, University of Évora, 7000-811 Évora, Portugal; 7Comprehensive Health Research Center, 7002-554 Évora, Portugal; 8Nursing School of Porto, 4200-072 Porto, Portugal; 9Group Inovation and Development in Nursing (NursID), Centro de Investigação em Tecnologias e Serviços de Saúde (CINTESIS), 4200-450 Porto, Portugal

**Keywords:** social media, internet addiction, mental health, higher education students, confinement

## Abstract

The use of social media was one of the most common way to keep in touch with friends and family during confinement. For higher education students, the fact that their universities were closed was a major change in their lives. The aim of this study is to relate the prevalence and type of social media with Internet addiction and mental health of university students in a district of Portugal during COVID-19 confinement. Mental health was studied by applying the reduced version of the Mental Health Inventory (MHI-5) and to measure the Internet use and dependence, the Internet Addiction Test (IAT) was used. The study (cross-sectional, descriptive, and correlational) used an online questionnaire, conducted on Google^®^ Forms and the link was sent to 4450 students, in the months of April to June 2020, during the confinement. A total of 329 valid questionnaires were obtained. We can conclude that regarding mental health, students in the 18–24 age group, single or divorced and who are not in a relationship, and with worse academic results, show worse levels of mental health. It is noteworthy that the students with the same characteristics also have the highest levels of addiction to internet.

## 1. Introduction

Due to the COVID-19 pandemic measures, to reduce the spread of the disease, were taken to implement social distancing such as home confinement, closing non-essential services and stores, banning crowd gatherings, and suspending face-to-face teaching in schools and universities with obvious challenges for students in higher education, for many of whom academic achievement is a factor of mental distress [1]. However, everyone had the need and desire to continue to communicate, and online social media networks were instrumental in maintaining that interaction.

Social media as web-based services allow individuals to (1) build a public or semi-public profile within a limited system, (2) articulate a list of other users with whom they share a connection, and (3) view and browse their list of connections and those made by others within the system. These are “communities” formed by public or semi-public profiles, where individuals can regulate who they connect with as well as browse the connections of others [2]. This social network integrates multiple levels of inter-individual social relationships and has direct associations with an individual’s health and well-being [3]. These online social networking sites are attracting increasing attention from researchers and the industry to learn about their capabilities and influences.

Many aspects of socialization depend on how individuals recognize each other. Understanding how, when, and whether individuals recognize others can provide fundamental insight into social relationships and behaviors [4]. Social relationships are likely to be affected by crisis in different ways, but they are an important channel of communication and social support and are most relevant in times of crisis. During the confinement caused by COVID-19 face-to-face interactions and face-to-face encounters were minimized and it is likely that individuals focused more on especially close, more meaningful, or more established relationships [5].

The emerging need for the use of social networking, information and communication technologies (ICT) has generated a number of research questions, related to their use and potential risk, but also potential for prevention or health promotion. Online social networking sites have become an important source of information for users, as well as a tool for social relationships [6]. What makes social networking sites unique is not that they allow individuals to meet strangers, but that they allow users to articulate and make visible their social media [2].

There are already several studies in this area of knowledge. From these studies some information can be drawn regarding aspects related to the social and educational characteristics of the participants and Internet addiction and the relationship with mental health. For example, Elmer in 2020 and Gómez-Salgado in 2022 found a relationship between mental health level and gender, finding that females during the COVID-19 pandemic had worse mental health indicators than males [7,8]. Other studies also reported that younger people also had lower mental health scores [9,10].

When it comes to the use of the internet and social media it is the male teenagers who are not in a love relationship that use them the most [11,12,13,14]. However, during the pandemic there was an increase in this use by women [15].

There does indeed seem to be a relationship between mental health level and social media. Individuals who have lower levels of MHI-5 are on the internet more, using social media more [16,17].

In a period when isolation has been imposed by public health conditions, the addition of the Internet and social media is an aspect of interest in mental health research. The time of cell phone use is a predictor of this type of addiction [18], but the addiction is not related to a specific type of social network [13].

In the research conducted by the authors, no studies were found that related social media use to Internet addiction and mental health of Portuguese higher education students during COVID-19 confinement.

In view of the above, the following research questions were formulated:(a)What is the type and frequency of social media use before and during confinement, by sex and personal characteristics of higher education students in a region of Portugal?(b)What is the association between internet addiction and social media use (type and frequency) during COVID-19 confinement of higher education students in one region of Portugal?(c)What is the association between higher education students’ mental health and social media use (type and frequency) during COVID-19 confinement in a region of Portugal?

## 2. Materials and Methods

### 2.1. Aims

The aim of this study is to relate the prevalence and type of social network use with internet addiction and mental health of higher education students in an Alentejo district in Portugal during the COVID-19 confinement.

### 2.2. Study Design

This cross-sectional, descriptive and correlational study used an online questionnaire run on the Google^®^ Forms applied to Portuguese higher education students. The study occurred from April to June 2020, during the COVID-19 confinement to identify the prevalence and type of social media use and its relationship with Internet addiction and students’ mental health. The Mental Health Inventory-5 (MHI-5), the Internet Addiction Test (IAT), and questions about personal characterization and social media use during confinement were the instruments of the study. The independent variables were mental health defined as “a state of well-being in which the individual realizes his or her own abilities, can cope with the normal stresses of life, can work productively and fruitfully, and is able to contribute to his or her community” (1), and internet addiction defined as “excessive or poorly controlled preoccupations, urges or behaviors regarding computer use and internet access that led to impairment or distress” (2). Dependent variables were defined by prevalence and typology of social media use during confinement.

### 2.3. Data Collection

A convenience sample, calculated for a margin of error of 5% and a confidence level of 90% was obtained. The questionnaire’s link was sent via e-mail to the 4450 students with active enrollment in the higher education institutions in a region of Portugal. A total of 329 valid questionnaires were obtained. In the first part of the questionnaire the study was described, and the participant could only continue to participate after giving consent. Ethical issues were always respected by the authors. Confidentiality of the data was assured, and study data were kept in the researchers’ personal databases, with security codes. The study was approved by the Ethics Committee of one of the institutions (Ethics Opinion n◦ SC/2020/316 of 20 February 2020) and by the Data Protection Officers of both institutions. The procedures were carried out in accordance with the Helsinki Declaration.

The sociodemographic characterization questions were developed by the research team and consists of self-report items such as: sex; age; marital status; type of love relationship; level of education; course; academic year; school performance rating.

The questionnaire to characterize social media use, before and during COVID-19, was also developed by the research team and consisted of questions about the use of social media such as Facebook, Twitter, LinkedIn, WhatsApp and Instagram. The questions addressed were, which social media do you use the most; how many years you have been using them; how many hours a day do you use them; where do you access most of the time (computer or mobile); and the main reasons for using them.

Mental health was studied by applying the reduced version of the Mental Health Inventory (MHI-5) translated and validated for Portugal by Pais-Ribeiro in 2011 [19]. Based on the thirty-eighth question Inventory, a reduced version called Mental Health Inventory 5 (MHI-5) was developed, composed of five items representing four dimensions of mental health: Anxiety, Depression, Loss of Emotional-Behavioral Control, and Psychological Well-Being [20]. The rating of the scale is obtained by summing the items (2 items with the rating reversed). Higher levels in the sum correspond to better mental health between 5 and 30.

To measure the Internet use and dependence, the Internet Addiction Test (IAT) was used, translated and validated for Portugal by Pontes and collaborators in 2014. It is rated using a six-point Likert scale. Assessing the Internet user’s involvement and classifying the addictive behavior into mild, moderate, and severe impairment. To obtain the total IAT score, the researcher only needed to sum the scores for each response provided by the participant, considering 0–30 = normal range; 31–49 = mildly addicted; 50–79 = moderately addicted; and 80–100 = severely addicted [21].

Prior to data collection, a pre-test of the questionnaire was administered to ten (10) students, who after its completion made no suggestions for changes, and the initial version was kept, as it was considered adjusted to the objectives and easy to understand and complete by the students invited to participate.

### 2.4. Statistical Analysis

Descriptive statistics (absolute and relative frequency, mean, and standard deviation) were used according to the type of variable to characterize the sample under study. The chi-square test was used to compare the proportions between the study variables and the demographic characteristics analyzed. To explore the correlations among the social media and the sociodemographic and academic results variables the spearman correlation was used.

Data analysis was performed using the computer program SPSS version 27 with a significance level of 5%.

## 3. Results

### Sociodemographic and Academic Characteristics and Mental Health and Internet Addiction

The sample included 329 participants from various master’s, undergraduate, and technology courses in one region of Portugal. The majority are between 18 and 24 years old (82.7%), single (63.3%), female (80.5%), attending an undergraduate course (83.3%), and 66.9% rate their academic results as good. Considering the characteristics of the sample and for the value of the MHI-5, students between 31 and 35 years of age (23.2), married or in a consensual union (22), men (21.1), and those who have an academic classification of good (20.2) present higher values. Those with the lowest values are women (19.45), students between the ages of 18 and 24 (19.47), those with no love relationship (19.54), and those with an academic rating of mediocre (14). For the IAT values, students between the ages of 18 and 24 (30.2), single/divorced (29.9), males (30.8), and those with a rating of mediocre (47.6) had the highest values. Those with the lowest IAT scores were women (29.1), students aged 31 to 35 (24.8), those who are married or cohabiting (23.6), and those who have a rating of very good (26.8). Overall, they have an average of 19.78 for the MHI-5 and 29.5 for the IAT (Table 1).

During confinement, males have a higher MHI-5 and IAT mean values than females, *p* = 0.003 and *p* = 0.428, respectively. Regarding age, there are no differences for the MHI-5 and IAT, *p* = 0.597 and *p* = 0.412, respectively. The analysis carried out considering marital/relational status also shows that there are no differences for the two variables MHI-5 (*p* = 0.350) and IAT (*p* = 0.779). The same happens for education level MHI-5 (*p* = 0.858) and IAT (*p* = 0.932) and for academic classification, respectively MHI-5 (*p* = 0.080) and IAT (*p* = 0.973).

In Table 2 the time of use of social media and main equipment to access it is analyzed by sex and MHI-5 and IAT.

It is possible to see that number of years using social media presents no statistically significant difference between man and women (*p* = 0.759). The same does not happen for the equipment used for accessing internet (*p* = 0.047) with women choosing mobile phones to do it more often than man. The mean MHI-5 value for women using mobile phone to access internet is statistically lower the men’s value (*p* = 0.009). For the IAT there are no statistically significant differences.

Table 3 presents the data for the sociodemographic information regarding internet addiction levels for our sample.

We found no statistical differences between men and women for the addiction levels (*p* = 0.705). Same for age groups (*p* = 0.679), marital/relational status (*p* = 0.513) and academic classification (*p* = 0.249). For level of education there is a statistically significant difference (*p* = 0.018) among the levels, with the distribution from lowest to highest addiction from post-graduation, Professional technical course, bachelor’s degree and master’s degree.

In Table 4, the values concerning the type of social media used during confinement due to COVID-19 and metal health, internet addiction and sex are presented.

We found no difference for sex in the use of Facebook (*p* = 0.478), LinkedIn (*p* = 0.965) and WhatsApp (*p* = 0.389). A statistically significant difference was found for Twitter, more men use it compared to women (*p* = 0.024) and Instagram (*p* = 0.025), used more by women. However, concerning MHI-5 (twitter *p* = 0.302; Instagram *p* = 0.632) and IAT (twitter *p* = 0.564; Instagram *p* = 0.418) there are no differences in the use of social media.

The use of the different social media by the entire sample, before and during confinement, is presented on Table 5.

The preferred social network, before and during confinement, was Instagram, followed by WhatsApp and Facebook. The least preferred social network was LinkedIn, used for professional networking.

The use during confinement of social media distributed by sociodemographic of the sample is presented in Table 6.

The mid-aged participants use more frequently, in a statistically significant way, Facebook than younger participants in this study (*p* = 0.005). By age group, 18–24 years old 62.86% used Facebook, 25 to 30 years old, 82.75% used Facebook, 31–35 years old 87.5%; 36–44 years old 85.71% and for those participants over 45 years old 66.6% used Facebook. For twitter 39.34% of men used it compared with 24.89% of women, this difference is statistically significant (*p* = 0.024). For twitter we found statistically significant differences for women using it more than men, younger participants more than older participants, single/no relationship more the marries/civil relationship, and the lower education level (technical degrees and bachelor’s degree) more than those in higher education levels (master and pos-graduation). There were not statistically differences for the different variables when studied for relations with the twitter use. Concerning WhatsApp only statistically significant differences were found for participants with higher academic classification using it more than others. For Instagram, statistically significant differences were found for, sex (women use it more), age (younger use it more), marital status (single/no relationship use more than others) and for course level (technical curses and bachelor’s use it more than pos-graduation and masters courses).

An aspect of interest is related to the time and reasons for the use of social media. Table 7 shows the relation between those aspects, before and during confinement, and mental health and internet addiction.

There are statistically significant differences for sex for the variable sharing life with other people (trips, photos, food, ...) before confinement (*p* = 0.039) with men more prevalent in this variable, and during confinement (*p* = 0.033) with women more prevalent. Additionally, women used social media, more than men, before the confinement (*p* = 0.023) for study and do group work with colleagues. No other variables presented statistically significant values. Most people used social media between 3 and 8 h a day. MHI-5 and IAT values were collected only during confinement.

## 4. Discussion

The present study found that women had lower MHI-5 scores than men during confinement. In a study by Elmer in 2020, female students had worse mental health when controlling for different levels of social integration during COVID-19 pandemic [7]. Same was mentioned by Gomez-Salgado that found women to have more mental health issues during the pandemic and also people that lived without a partner [8]. In the present study, people married or in a civil union had better MHI-5 scores, and the worst scores were for people that were not in a relationship. Age is also a factor when looking at mental health, our study found that the young adults (18–24 years) had worse mental health, same was found in a systematic review that states that younger people are at risk of experience more mental health problems during pandemic times [9,10] and students closer to graduating, similar age group to our study, faced increases in anxiety, feeling of loneliness, and depression [1]. Academic achievement was also related to mental health, the people with higher academic grades had better mental health scores that those with reported lower academic grades. Academic achievement is described in the literature as factor for mental distress [22].

Various individual and socio-economic factors have been found to contribute to students’ academic success. Many college students had their universities abruptly closed during COVID-19, with large reductions in social interactions and likely experienced increases in loneliness, the latter being associated with social network use [11]. Extensive research suggests that loneliness is the cause of increased social media use and not that social media use is the cause of loneliness [12].

Social media created informally within learning communities can further influence student learning outcomes. However, the impact of these social media is difficult to measure and quantify because social media are multidimensional and dynamic in educational contexts [5]. In our study men presented a higher IAT value than women, same for age group from 18 to 24 years old, single or divorced with low academic grades. Studies found that men and adolescents are a high-risk group for dangerous, excessive, or impulsive internet use, leading to negative life outcomes [13,14], but internet addiction is not associated with a specific social network or set of social media [13]. Our study found no relations between mental health evaluated by MHI-5 and internet addiction evaluated by IAT, otherwise is described in the literature stating that there is an association between internet addiction with mental health [23,24]. Additionally, also, a high prevalence of mental health problems, positively associated with frequently social media exposure during the COVID-19 outbreak [16].

The COVID-19 pandemic has been associated with negative psychological outcomes such as increased rates of depression, anxiety and stress among the general population [25,26], which has been offset by social media as a result of social exclusion [11]. Some studies show positive associations between time spent on social media and negative mental health outcomes (depression and anxiety) [17]. The present study found no differences between men and women in the time spent on social media. Social distancing and isolation at home has altered the use of social media, making their use one of the most common forms of social interaction [11]. This communication among people is also described in the literature, in many social media, participants do not necessarily “socialize” or seek to meet new people; instead, they mostly interact with people who are already part of their larger social network [2]. Intrinsic motivation to use social media solely for emotional reward leads to high engagement with it, while time spent on social media throughout the day leads to habitual use, which in some contexts amounts to automatic use due to certain learned associations [27].

In our study women use the mobile phones to gain access to social media more than men do and prefer Instagram. Men prefer twitter. Before the pandemic men used social media to share their life, during pandemic it was women who share more their lives on social media. The literature has shown that female reported a higher increase in the engagement in social media, information research, and video streaming than males. Additionally, revealed an increased usage of all online applications during the lockdown [28]. Women and nonbinary respondents were twice as likely than men to pick social media for coping [15]. Another study found that spent on smartphones was amongst the strongest predictors of social media addiction [18].

Literature states that there was an increase in daily time spent on social media and their dependence during confinement compared to usual social media use at pre-COVID-19. In our study the preferred social media, before and during confinement were, in order of preference, Instagram, WhatsApp and Facebook. Same engagement with social media was found to not change significantly during COVID-19 [11]. A study conducted by Pop and colleagues in 2021 among Romanian university students found that although participants reported using several social media, they used Facebook the most, followed by Instagram, WhatsApp, TikTok and YouTube, and students spent on average 4.81 ± 3.60 h per day on these networks. The most common reasons students used social media were for socializing, entertainment, information or relaxation [29]. In the present study, the most used social media are Instagram, WhatsApp and Facebook. According to another survey conducted by AlFaris in 2018, 97% of respondents use social network. YouTube, WhatsApp and Twitter were the most popular in that study. In terms of general use, male students are significantly more likely to visit YouTube and Facebook, while the opposite is true for Instagram. In our study statistically significant differences were found for sex in the social media use in Instagram where women used it more than men, same in twitter. About 71% visit social media more than 4 times a day and 55% spend 1–4 h a day there. The main reasons for using social media are entertainment, news and communication, research. No significant relationship was found between average academic level and frequency of social media use or daily use in the classroom. Although almost all students used social media, only a minority did so for academic purposes. There was no relationship between social media use and academic performance [30]. Our study found that the students with better academic results are the ones using more WhatsApp that the others in a statistically significant way.

In our study most of the participants used social media between 3 to 8 h a day, with the time spent on social media increasing during confinement. In the literature is possible to find that women accessed social networking sites between 6–10 h a day, more than men who spent between 1–5 h on social networking sites every day [31]. These are really high number of hours spent online, however there is no evidence that time spent using social media might influence an individual’s mental health over time [32]. In 2022, young adults use the following social media, in descending order, TikTok, Instagram, Snapchat, Facebook, Twitter, Twitch and WhatsApp. In our study, Twitter and Instagram are the social media preferred by young adults [33]. Regarding marital status, literature states that social media are a stressor for relationships and can create conflict between individuals [34]. As in our study, other study found that social media use is associated with individuals who are not in a relationship (single/divorced) and may even have low relationship commitment [35].

## 5. Conclusions

Our study found better MHI-5 values for the 31–35 age group, for participants who are married/in a consensual union, male and with good academic achievement. The worst MHI-5 values were found for females, the 18–24 age group and participants who are not in a relationship, those with worse academic achievement also have worse MHI-5 values.

For IAT, those who are most addicted are men, participants aged 18–24, single or divorced, and with poorer academic achievement. The least addicted are women and participants in the age group 31–35, married or in a consensual union and with an academic score of very good.

It can be concluded that participants in the age group 18–24 years have the worst MHI-5 score and more IAT score. Participants who are not in a relationship and who are single or divorced have worse mental health and more internet use addiction, the same is true for participants who have a worse academic score. There were no differences in MHI-5 but for IAT scores, participant’s in master’s level were more addicted to internet.

No differences were found for the number of years of use of social media between men and women. Women give preference to cell phones to access social media, preferring Instagram and twitter, same for younger people. Older participants prefer Facebook. Technical Courses and bachelor’s degree students use more Instagram and Twitter, the same for single/divorces people. Before the pandemic men used social media to share their life experiences, with the confinement this pattern changed and now women use social media for this purpose.

This study presents results from a specific population of a district of Portugal and higher education students, and also the sample is not representative in terms of the gender of the respondents, which may be a weakness in the results obtained. We consider it important to extend this study to other regions of the country and to population groups that also suffered the constraints of confinement, since the issue of social media use is a key aspect in studying the mental health of young people, particularly higher education students in such a challenging period of their personal and academic lives. The fact that there has been confinement with consequent isolation are certainly determining factors for mental health, internet use, and potential addictive social media behavior. Since these have become one of the main, if not the main, engine for relationships with peers, family and new contacts. The study of this period and its future consequences is critical to understanding the implications for social, academic and social media use behaviors in the years to come.

## Figures and Tables

**Table 1 ijerph-20-00664-t001:** Sociodemographic, academic characterization, Mental Health and Internet Addiction indexes of the sample.

Sociodemographic and Academic Characteristics		n	%	MHI-5	IAT
Sex	Female	265	80.5	19.45	29.17
	Male	64	19.5	21.17	30.82
	Total	329	100	19.78	29.5
Age	18 a 24	272	82.7	19.47	30.18
	25 a 30	29	8.8	20.62	26.4
	31 a 35	8	2.4	23.12	24.87
	36 a 44	14	4.3	21.07	25.07
	>44	6	1.8	22.5	29.00
Marital/relational status	Married/civil union	20	6.1	22.0	23.68
	Single/divorced	208	63.3	19.69	29.88
	No relationship	101	30.7	19.54	29.81
Level of education	Professional Technical Course	40	12.2	21.22	23.66
	Bachelor’s Degree	274	83.3	19.54	30.30
	Master’s Degree	12	3.6	19.50	31.25
	Post-Graduation	3	0.9	24.00	25.66
Academic classification	Mediocre	4	1.2	14	47.66
	Adequate	59	17.9	18.45	31.42
	Good	220	66.9	20.2	29.28
	Very Good	46	14.00	19.97	26.84

**Table 2 ijerph-20-00664-t002:** Time of use of social media and main equipment of use, by sex and MHI-5 and IAT.

Time of Use of Social Media and Main Equipment of Use, by Sex and Mental Health and Internet Addiction		Female n = 265				Male n = 64			
		n	%	MHI-5	IAT	n	%	MHI-5	IAT
How many years have you been using social Networks?	Less than 1 year	1	0.38	19	21	0	0	0	0
	1–5 years	26	9.81	19	26.8	8	12.5	20.5	28.7
	6–10 years	160	60.38	19.3	29.1	37	57.81	20.9	31.2
	More than 10 years	78	29.43	20	29.8	16	25	22.4	28.6
	Not applicable	0	0	0	0	3	4.69	19.3	37
Where do you access social media most often?	Mobile phone	258	97.36	19.4	29	58	90.62	21	30
	Computer or tablet	7	2.64	19.7	33.7	3	4.69	25.3	39.3
	Not applicable	0	0	0	0	3	4.69	19.3	37

**Table 3 ijerph-20-00664-t003:** Sociodemographic of the sample and the values of Internet Addiction Test.

		Internet Addiction Test							
Sociodemographic and Academic Characteristics		0–30 Normally Addicted		31–49 Mildly Addicted		50–79 Moderately Addicted		80–100 Severely Addicted	
		N	%	n	%	n	%	n	%
Sex	Female n = 265	143	53%	109	41%	13	4.9%	0	0%
	Male n = 64	35	54%	21	32.8%	8	1.2%	0	0%
	Total n = 329	178	54%	130	39.5%	21	6.3%	0	0%
Age	18 a 24 n = 272	146	53.6%	108	39.7%	18	6.6%	0	0%
	25 a 30 n = 29	15	51.7%	12	41.3%	2	6.8%	0	0%
	31 a 35 n = 8	4	50%	4	50%	1	12.5%	0	0%
	36 a 44 n = 14	9	62.5%	4	28.5%	0	0%	0	0%
	>44 n = 6	4	66.6%	2	33.3%	0	0%	0	0%
Marital/Relational status	Married/civil Union n = 20	12	60%	7	35%	1	5%	0	0%
	Single/divorced n = 208	111	52.8%	88	42.3%	9	4.3%	0	0%
	No relationship n = 101	55	54.4%	35	34.6%	11	10.8%	0	0%
Level of education	Professional Technical Course n = 40	19	47.5%	18	45%	3	7.5%	0	0%
	Bachelor’s Degree n = 274	150	54.7%	106	38.6%	18	6.5%	0	0%
	Master’s Degree n = 12	7	58.3%	5	41.6%	0	0%	0	0%
	Post-Graduation n = 3	2	66.6%	1	33.3%	0	0%	0	0%
Academic classification	Mediocre n = 4	2	50%	2	50%	0	0%	0	0%
	Adequate n = 59	37	62.7%	17	28.8%	14	23.7%	0	0%
	Good n = 220	120	54.5%	86	39%	2	9%	0	0%
	Very Good n = 46	19	41.3%	25	54.3%	0	0%	0	0%

**Table 4 ijerph-20-00664-t004:** Type of social media during confinement COVID-19 and Mental Health, Internet Addiction and sex.

Use of Social Networks		Sex	n	%	MHI-5	IAT
Facebook	Did not use	Female	91	34.3	18.8	29.6
		Male	20	31.2	22.7	34.2
	Use	Female	174	64.9	19.8	28.8
		Male	43	67.1	20.4	28.7
Twitter	Did not use	Female	182	68.6	19.2	28.3
		Male	35	54.6	21.6	30
	Use	Female	63	23.7	20.2	31.4
		Male	21	32.8	20.3	31.3
LinkedIn	Did not use	Female	258	97.3	19.5	19.1
		Male	60	93.7	21.1	30.5
	Use	Female	7	2.6	16.8	28.1
		Male	3	4.6	22.6	29.3
WhatsApp	Did not use	Female	51	19.2	18.9	28.6
		Male	11	17.1	22.5	33.3
	Use	Female	214	80.7	19.5	29.2
		Male	52	81.2	20.8	29.7
Instagram	Did not use	Female	38	14.3	18.8	26.6
		Male	13	20.3	22.1	33.7
	Use	Female	227	85.6	19.5	29.5
		Male	50	78.1	20.9	29.6

**Table 5 ijerph-20-00664-t005:** Type of social media used before and during confinement COVID-19 by the study participants.

Use of Social Media		Before n	Before %	During n	During %
Facebook	Did not use	124	37.7	111	33.7
	Use	205	62.3	218	66.3
Twitter	Did not use	260	79.0	218	66.3
	Use	69	21.0	84	25.5
LinkedIn	Did not use	323	98.2	319	97.0
	Use	6	1.8	10	3.0
WhatsApp	Did not use	58	17.6	64	19.5
	Use	271	82.4	265	80.5
Instagram	Did not use	49	14.9	52	15.8
	Use	280	85.1	277	84.2

**Table 6 ijerph-20-00664-t006:** Sociodemographic of the sample and the use during confinement, of social media.

Sociodemographic and Academic Characteristics		Social Network Use during Confinement									
		Facebook		Twitter		LinkedIn		WhatsApp		Instagram	
		n	Correlation (*p*-Value)	n	Correlation (*p*-Value)	n	Correlation (*p*-Value)	n	Correlation (*p*-Value)	n	Correlation (*p*-Value)
Sex	Female	178	ρ = 0.039 (*p* = 0.480)	60	ρ = −0.129 (*p* = 0.024) *	8	ρ = −0.002 (*p* = 0.965)	211	ρ = −0.048 (*p* = 0.390)	229	ρ = 0.124 (*p* = 0.025) *
	Male	40		24		2		54		48	
Age	18 a 24	171	ρ = 0.047 (*p* = 0.005) *	77	ρ = −0.154 (*p* = 0.007) *	7	ρ = 0.058 (*p* = 0.298)	215	ρ = 0.087 (*p* = 0.113)	236	ρ = −0.174 (*p* = 0.002) *
	25 a 30	24		6		1		24		26	
	31 a 35	7		0		2		7		5	
	36 a 44	12		1		0		14		7	
	>44	4		0		0		5		3	
Marital/Relational status	Married/civil Union	18	ρ = −0.079 (*p* = 0.151)	1	ρ = 0.131 (*p* = 0.023) *	1	ρ = −0.012 (*p* = 0.825)	19	ρ = 0.022 (*p* = 0.687)	11	ρ = 0.116 (*p* = 0.035) *
	Single/divorced	136		51		6		161		178	
	No relationship	64		32		3		85		88	
Level of education	Professional Technical Course	27	ρ = −0.021 (*p* = 0.700)	15	ρ = −0.115 (*p* = 0.046) *	1	ρ = 0.032 (*p* = 0.564)	38	ρ = −0.096 (*p* = 0.081)	34	ρ = −0.138 (*p* = 0.012) *
	Bachelor’s Degree	182		67		8		214		237	
	Master’s Degree	6		2		1		10		6	
	Post-Graduation	3		0		0		3		0	
Academic classification	Mediocre	3	ρ = −0.010 (*p* = 0.863)	2	ρ = −0.079 (*p* = 0.168)	0	ρ = −0.012 (*p* = 0.825)	4	ρ = −0.112 (*p* = 0.042) *	3	ρ = −0.024 (*p* = 0.667)
	Adequate	40		17		1		51		52	
	Good	144		56		9		177		183	
	Very Good	31		9		0		33		39	

* Statistically significant.

**Table 7 ijerph-20-00664-t007:** Time and reasons for the use of social media before and during confinement COVID-19 and Mental Health and Internet Addiction.

Time and Reasons for the Use of Social Media		Female n = 265				Male n = 64			
		n	%	MHI-5	IAT	n	%	MHI-5	IAT
How many hours did you use social media before the confinement?	No	2	0.7			2	3		
	Less than 1	21	7.9			2	3		
	1 to 2	84	31.6			20	31.2		
	3 to 5	114	43			28	43.7		
	6 to 8	33	12.4			7	10.9		
	Over 8	11	4.1			5	7.8		
How many hours did you use social media during the confinement?	No	1	0.3	22	43	0	0		
	Less than 1	9	3.3	15.2	14.8	2	3	25.5	36
	1 to 2	39	14.7	19	24.2	9	14	19.7	29.3
	3 to 5	94	35.4	19.7	27.7	17	26.5	22.5	27.1
	6 to 8	71	26.7	19.6	30.6	23	35.9	21.4	29.8
	Over 8	51	19.2	19.7	35.6	13	20.3	19.3	36.2
Main reasons why you accessed social media before confinement?	Contact with family or friends	225	84.9			53	82.8		
	Work	63	23.7			13	20.3		
	Meet other people or make new friends	17	6.4			7	10.9		
	Play	40	15			16	25		
	Sharing life with other people (trips, photos, food, etc.)	81	30.5			21	32.8		
	Get to know other people’s lives (travels, photos, food, etc.)	71	26.7			17	26.5		
	Obtain and/or share curiosities, news or information	160	60.3			34	53.1		
	Study and do group work with colleagues	170	64.1			40	62.5		
What are the main reasons why you accessed social media during confinement?	Contact with family or friends	238	89.8	19.8	29.4	55	85.9	20.8	31
	Work	70	26.4	19.7	25.4	18	28.1	20.7	27.5
	Meet other people or make new friends	18	6.7	19.4	37.1	6	9.3	19.6	29.6
	Play	62	23.3	20.4	32.3	17	26.5	20.8	33.8
	Sharing life with other people (trips, photos, food, etc.)	71	26.7	19.6	31.8	15	23.4	20.8	31.6
	Get to know other people’s lives (travels, photos, food, etc.)	79	28.9	19.5	32.5	18	28.1	21.8	35.5
	Obtain and/or share curiosities, news or information	180	67.9	19.5	27.8	38	59.3	22.6	31.5
	Study and do group work with colleagues	174	65.6	19.5	29.5	45	70.3	20.5	28.1

## Data Availability

Data available on request due to ethical restrictions.

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
