# Peer review of "Social Media Use and Its Association with Mental Health and Internet Addiction among Portuguese Higher Education Students during COVID-19 Confinement"

_ijerph, 2022, doi:10.3390/ijerph20010664_

Round 1

Reviewer 1 Report

Thank you for the opportunity to review the article “Social networks use and its association with mental health and internet addiction of Portuguese higher education students during COVID-19 confinement”. The article is engaging, exploring the prevalence of Internet addiction and mental health of university students from a district in Portugal when using social networks during the COVID-19 pandemic.

The theoretical background is appropriate for this subject. Still, I would have preferred it to be more extensive - some of the papers outlined in the discussion part could be in this first part, and the discussion section should focus more on the significance of the results.

Overall, the subject is correctly attributed to the Mental Health section of the International Journal of Environmental Research and Public Health journal, special issue “Addictive Behaviors and Mental Health in Adolescents and Young Adults,” by studying a contemporary situation and its impact on a local level. 

It is important to emphasize that the article is well-structured, and the results are clearly presented. 

However, there are just a few suggestions that the authors could introduce to improve the overall quality of the article.  This is related to the limits of the research - limitations, and constraints should be presented.

Author Response

Dear Reviewer 1,

The authors appreciate and recognize the work done by the reviewer 1 and express their gratitude for the comments.

The theoretical background is appropriate for this subject. Still, I would have preferred it to be more extensive - some of the papers outlined in the discussion part could be in this first part, and the discussion section should focus more on the significance of the results.

Answer: The theoretical background was updated, and a few paragraphs were added with information on papers outlined in the discussion, lines 41-42 and 70-86. Where is possible to read:

“…schools and universities with obvious challenges for students in higher education, for many of whom academic achievement is a factor of mental distress [1]. But everyone had…”

And

“There are already several studies in this area of knowledge. From these studies some information can be drawn regarding aspects related to the social and educational characteristics of the participants and Internet addiction and the relationship with mental health.  For example, Elmer in 2020 and Gómez-Salgado in 2022 found a relationship between mental health level and gender, finding that females during the COVID-19 pan-demic had worse mental health indicators than males [7, 8]. Other studies also reported that younger people also had lower mental health scores [9, 10].

When it comes to the use of the internet and social media it is the male teenagers who are not in a love relationship that use them the most [11-14].  However, during the pan-demic there was an increase in this use by women [15].

There does indeed seem to be a relationship between mental health level and social networks. Individuals who have lower levels of MHI-5 are on the internet more, using social media more [16, 17].

In a period when isolation has been imposed by public health conditions, the addition of the Internet and social networks is an aspect of interest in mental health research.  The time of cell phone use is a predictor of this type of addiction [18], but the addiction is not related to a specific type of social network [13].”

However, there are just a few suggestions that the authors could introduce to improve the overall quality of the article.  This is related to the limits of the research - limitations, and constraints should be presented.

Answer: More information was added to that statement of limitations and constraints namely from line 380-383, with more elaborated comments on what was already written on this topic (lines 377-380), now it reads:

This study presents results from a specific population of a district of Portugal and higher education students, and also the sample is not representative in terms of the gender of the respondents, which may be a weakness in the results obtained. We consider it important to extend this study to other regions of the country and to population groups that also suffered the constraints of confinement, since the issue of social media use is a key aspect in studying the mental health of young people, particularly higher education students in such a challenging period of their personal and academic lives.

Reviewer 2 Report

The paper presented defines a very significant subject and is applied, as the authors point out, in a geographical zone for which there are no data on the field. In my opinion, the article answers the research questions formulated and the described methodology is correctly applied. Even so, the article lacks any relevant results that could contribute to the global discussion on the impact of social media on the mental health of young people. Perhaps the following comments can help to find that relevance that is not perceived from the text:

Although the article focuses solely on Portugal, there are numerous studies that relate the use of social media with addiction and mental health problems in different countries and that are not presented as background to the thematic presentation. At the same time, hundreds of problems have been identified that can also be complemented by previous studies that analyze the impact of social media on young people and their health, not just mental health.

In the limitations of the study it should be mentioned that the sample is not representative in terms of the gender of the respondents.

The results in table five are truncated in the last two columns (n - %).

Regarding the results and discussion, the article correctly applies the proposed methodology and indicates relevant indicators on the probability of having or suffering greater mental health problems when using social networks in relation to some variables analyzed.

Although it would be remarkable to verify these relationships with other types of research techniques, we understand that the method used is applied correctly and, with it, obtained results solve the research questions. Even so, novel results are not highlighted, which, perhaps, can be highlighted in the conclusions.

The conclusions and the discussion of results are very similar. The conclusions can provide a broader perspective of the problem of social media on mental health and the generation of addiction in young people.

Author Response

Dear Reviewer 2,

The authors appreciate and recognize the work done by the reviewer 2 and express their gratitude for the comments.

…there are numerous studies that relate the use of social media with addiction and mental health problems in different countries and that are not presented as background to the thematic presentation. At the same time, hundreds of problems have been identified that can also be complemented by previous studies that analyze the impact of social media on young people and their health, not just mental health.

Answer: The theoretical background was updated, and a few paragraphs were added with information on papers outlined in the discussion, lines 41-42 and 70-86. Where is possible to read:

“…schools and universities with obvious challenges for students in higher education, for many of whom academic achievement is a factor of mental distress [1]. But everyone had…”

And

“There are already several studies in this area of knowledge. From these studies some information can be drawn regarding aspects related to the social and educational characteristics of the participants and Internet addiction and the relationship with mental health.  For example, Elmer in 2020 and Gómez-Salgado in 2022 found a relationship between mental health level and gender, finding that females during the COVID-19 pan-demic had worse mental health indicators than males [7, 8]. Other studies also reported that younger people also had lower mental health scores [9, 10].

When it comes to the use of the internet and social media it is the male teenagers who are not in a love relationship that use them the most [11-14].  However, during the pan-demic there was an increase in this use by women [15].

There does indeed seem to be a relationship between mental health level and social networks. Individuals who have lower levels of MHI-5 are on the internet more, using social media more [16, 17].

In a period when isolation has been imposed by public health conditions, the addition of the Internet and social networks is an aspect of interest in mental health research.  The time of cell phone use is a predictor of this type of addiction [18], but the addiction is not related to a specific type of social network [13].”

In the limitations of the study it should be mentioned that the sample is not representative in terms of the gender of the respondents.

Answer: This information was added and in lines 377-379 it now reads: 

This study presents results from a specific population of a district of Portugal and higher education students, and also the sample is not representative in terms of the gender of the respondents, which may be a weakness in the results obtained.

The results in table five are truncated in the last two columns (n - %).

Answer: the table is now correct

Although it would be remarkable to verify these relationships with other types of research techniques, we understand that the method used is applied correctly and, with it, obtained results solve the research questions. Even so, novel results are not highlighted, which, perhaps, can be highlighted in the conclusions.

The conclusions and the discussion of results are very similar. The conclusions can provide a broader perspective of the problem of social media on mental health and the generation of addiction in young people.

Answer: The comments were considered, and the conclusion was changed, it now reads,  in lines 357-389:

Our study found better MHI-5 values for the 31-35 age group, for participants who are married/in a consensual union, male and with good academic achievement. The worst MHI-5 values were found for females, the 18-24 age group and participants who are not in a relationship, those with worse academic achievement also have worse MHI-5 values.

For IAT, those who are most addicted are men, participants aged 18-24, single or divorced, and with poorer academic achievement. The least addicted are women and participants in the age group 31-35, married or in a consensual union and with an academic score of very good.  

It can be concluded that participants in the age group 18-24 years have the worst MHI-5 score and more IAT score. Participants who are not in a relationship and who are single or divorced have worse mental health and more internet use addiction, the same is true for participants who have a worse academic score. There were no differences in MHI-5 but for IAT scores, participant’s in master’s level were more addicted to internet.

No differences were found for the number of years of use of social networks between men and women.  Women give preference to cell phones to access social media, prefer-ring Instagram and twitter, same for younger people. Older participants prefer Facebook. Technical Courses and bachelor’s degree students use more Instagram and Twitter, the same for single/divorces people. Before the pandemic men used social networks to share their life experiences, with the confinement this pattern changed and now women use social networks for this purpose.

This study presents results from a specific population of a district of Portugal and higher education students, and also the sample is not representative in terms of the gender of the respondents, which may be a weakness in the results obtained. We consider it important to extend this study to other regions of the country and to population groups that also suffered the constraints of confinement, since the issue of social media use is a key aspect in studying the mental health of young people, particularly higher education students in such a challenging period of their personal and academic lives. The fact that there has been confinement with consequent isolation are certainly determining factors for mental health, internet use, and potential addictive social media behavior. Since these have become one of the main, if not the main, engine for relationships with peers, family and new contacts. The study of this period and its future consequences is critical to understanding the implications for social, academic and social media use behaviors in the years to come. 
